

# Live software documentation of design pattern instances

Filipe Lemos[1], Filipe F. Correia[1,2], Ademar Aguiar[1,2] and Paulo G. G. Queiroz[2,3]

[1] Faculty of Engineering, University of Porto, Porto, Portugal
[2] INESC TEC, Porto, Portugal
[3] Universidade Federal Rural do Semi-Árido, Mossoró, RN, Brazil

## ABSTRACT

**Background:** Approaches to documenting the software patterns of a system can support intentionally and manually documenting them or automatically extracting them from the source code. Some of the approaches that we review do not maintain proximity between code and documentation. Others do not update the documentation after the code is changed. All of them present a low level of *liveness*.
**Approach:** This work proposes an approach to improve the understandability of a software system by documenting the design patterns it uses. We regard the *creation* and the *documentation* of software as part of the same process and attempt to streamline the two activities. We achieve this by increasing the feedback about the pattern instances present in the code, during development—*i.e.*, by increasing *liveness*. Moreover, our approach maintains proximity between code and documentation and allows us to visualize the pattern instances under the same environment. We developed a prototype—*DesignPatternDoc*—for IntelliJ IDEA that continuously identifies pattern instances in the code, suggests them to the developer, generates the respective pattern-instance documentation, and enables live editing and visualization of that documentation.
**Results:** To evaluate this approach, we conducted a controlled experiment with 21 novice developers. We asked participants to complete three tasks that involved *understanding* and *evolving* small software systems—up to six classes and 100 lines of code—and recorded the duration and the number of context switches. The results show that our approach helps developers spend less time understanding and documenting a software system when compared to using tools with a lower degree of liveness. Additionally, embedding documentation in the IDE and maintaining it close to the source code reduces context switching significantly.

# INTRODUCTION

Throughout the life cycle of a software project, developers often record knowledge as different software artifacts, which can help understand what has been done, how the system works, and why it was made in such a way. This information is important for new project team members but also for the original developers, who may lose some of their knowledge, making documentation particularly useful when trying to reconnect with a part of the system. Without a clear notion of how the software works internally, developers

Corresponding author
Filipe F. Correia,
filipe.correia@fe.up.pt

may break existing designs when adding new features or modifying existing ones. Documentation can also support reuse, given that to reuse a piece of software, we first have to understand and learn how it works (*Aguiar, 2003*). Furthermore, successful software keeps evolving to meet its changing requirements, and to keep its full value, its artifacts need to be updated when new knowledge is acquired. It is often a challenge to keep all these artifacts consistent—*i.e.*, to make sure that two related artifacts do not express different ideas (*Correia, 2010*, *2015*).

These issues span several types of documentation artifacts. Use cases, class diagrams, and other models help describe software systems' functionality and architectural and design details. Source code also plays a part in preserving knowledge, as developers often strive for *self-documenting code* (*Spinellis, 2010*).

Designing software has been an early concern of software developers (*Parnas, 1972*), gained renewed interest and research avenues with *design patterns* (*Gamma et al., 1995*) and is ingrained in the activity of professionals, and a subject of research (*Tang et al., 2010*; *Farshidi, Jansen & van der Werf, 2020*; *Sousa, Ferreira & Correia, 2021*; *Riehle, Harutyunyan & Barcomb, 2021*). Therefore, in this work, we focus specifically on documenting *software design*. Some of the early results of this research are part of Filipe Lemos' masters thesis: https://hdl.handle.net/10216/128568.

## Documenting object-oriented design

Design patterns offer reusable solutions to recurring design problems and are particularly useful when designing and describing complex object-oriented software systems. A good way to document software is by describing the *pattern instances* that compose it. According to *Odenthal & Quibeldey-Cirkel (1997)*, good practices for documenting pattern instances include providing an overview of the design context, highlighting the reason why the pattern was instantiated, and describing the design in detail. This description must comprise outlining the pattern's participant roles, illustrating the possible interactions between them, providing the benefits and consequences of using the pattern and, finally, identifying special features of its implementation by referencing the source code. Due to the widely-known vocabulary established by patterns, using them to document software systems can make it easier to share knowledge and design experience with other developers.

## Programming *vs.* documenting

In their practice, software developers regularly have to switch from *programming* to *documenting*. These two activities are firmly related to each other since the output of each one serves as input to the other, creating a mutual feedback loop. This loop induces a constant context-switching, which may impede smooth flow (*Power & Conboy, 2014*) or lead to the loss of important knowledge. To avoid context-switching, developers may find themselves constantly postponing one of these two activities, often the *documenting* one, to remain longer in the same context. This may bring on inconsistencies since one of the artifacts is not evolving with the other. We may try to sync them back together later but, the more we increase the volume of pending changes, the harder it gets to ensure the artifacts become in-sync again. This can have a critical impact on the reuse and the ability

to understand the software since we will be dealing with outdated or inconsistent documentation.

### *Liveness* in software development

To address the duality between *programming* and *documenting* in this work, we rely on the notion of *Live Software Development* (LiveSD) (*Aguiar et al., 2019*). LiveSD consists of the extension of *Live Programming* to the entire software development life-cycle. While Live Programming promotes a constant awareness of the program state to help to reduce the edit-compile-run cycle, LiveSD aims to go beyond *programming*, as a way to support *technical agility*—the application of the *inspect and adapt* principle used in agile methods to technical software development activities.

*Tanimoto (2013)* defined liveness as a set of behaviors by a programming environment that provide information to programmers about what they are constructing. Additionally, Tanimoto proposes six levels to define liveness: (1) Informative; (2) Directly requested feedback; (3) Delayed non requested feedback; (4) Instantaneous feedback; (5) Predictive and suggestive feedback and (6) Strategically predictive.

Level 1 is merely informative; in our specific context it can consist of the description of which design pattern instances exist in a given codebase, in text format, or using a UML diagram. Level 2 assumes an executable artifact, allowing the developer to receive feedback upon request. For instance, the developer may press a button to show an informative box listing a set of design pattern instances extracted from the source code. In level 3, the feedback is provided sometime after an event (delayed automatic response). This feedback may, for example, warn the developer that some method is missing in a pattern instance, a few seconds after saving the program, without direct request by the developer. Level 4 provides feedback in real-time. The feedback is not requested by the developer but can be triggered by an event. For example, running the program in the background to search for design pattern instances and, after a few seconds without typing, results are displayed on the screen. In level 5, while running the program in the background, the system predicts future actions and suggests them to the developer. Finally, in level 6, systems predict future actions in the large, for a large unit of software.

Increasing *liveness* in the context of documentation of pattern instances is part of our main research objectives, as described in the next section.

## RESEARCH STRATEGY

The purpose of this work is to streamline the process that comprises the *creation* and the *documentation* of software, making it easier to switch between the activity of *programming*—in particular, instantiating design patterns, and the activity of *documenting*—in particular, describing its design pattern instances.

### Research questions

Our research is based on the hypothesis that *by increasing the level of liveness of documenting pattern instances, we will streamline the process of switching between*

*programming and documenting, making it easier to understand a system's design and to document it.*

In other words, we equate an increase of liveness to a higher immediacy in the awareness regarding the relationship between the code and its pattern-instance documentation, within the same environment, and we expect that it will: (a) allow developers to understand a software system design more easily, as developers will be exposed to the design documentation when reading or writing source code; (b) allow developers to keep design documentation up to date more easily, as they will quickly gain awareness when code and documentation drift apart; and (c) reduce the overall amount of context switching between coding and documenting activities, supported by a constant awareness on the code and its respective documentation. Therefore, to evaluate and discuss our hypothesis we formulate the following research questions:

- **RQ1.** *To what extent can liveness make it easier to understand a software system's design in terms of its pattern instances?*
- **RQ2.** *To what extent can liveness make it easier to keep software documentation updated?*
- **RQ3.** *To what extent can liveness reduce context switching between programming and documenting?*

To answer these questions, we consider an approach to document pattern instances that seeks to reduce the length of the feedback loops between *programming* and *documenting*. We build this work around the ideas of providing awareness towards existing documentation during *programming* activities and providing awareness towards the existing implementation during *documenting* activities.

## Methodology

This research was conducted according to the three stages detailed below.

### Literature review

We first review existing approaches that may support our vision. In particular, we compare them and highlight the extent to which they support liveness and good practices for keeping software documentation consistent with the source code (*cf.* "Related Work"). The works that we analyze were collected through a literature review that follows precepts of a systematic literature review (*Kitchenham, 2004*). The bulk of our search was carried out between October 2019 and January 2020, and further completed in October 2023 (*cf.* "Detecting Pattern Instances in Code"). We used IEEE Xplore and Google Scholar databases and focused on approaches and tools for the documentation and detection of design patterns, using the terms "documenting", "design patterns" and "detection" to narrow down the number of results. The articles analyzed met the following inclusion criteria: presented approaches to document or detect design patterns; or presented tools to document or detect design patterns on source code. Articles that met the following exclusion criteria were not explored further: presented approaches or tools to document solutions that are not design patterns; or were not written in English.

### Approach implementation

We use these good practices and the notion of liveness to conceive *DesignPatternDoc*, an IDE plugin that supports pattern-based software documentation (*cf.* "The *Designpatterndoc* Plugin"). The plugin provides the fourth level of liveness for each activity in relation to the other—*programming* and *documenting*. Namely, this tool (a) identifies pattern instances in the source code and suggests, in real-time, how to document them, and (b) supports editing and visualizing pattern-instance documentation during programming.

### Controlled experiment

We used the *DesignPatternDoc* plugin to run a controlled experiment with users and generate insights regarding our research questions. The controlled experiment follows the process described by *Wohlin et al. (2012)* and is divided into five main activities: definition, planning, operation, analysis and interpretation, and finally, presentation.

In summary, we divided the participants into *control* and *experimental* groups. The latter had access to our tool. We asked both groups to complete a set of software development tasks, which involved understanding and evolving a software system. During the experiment, we measured the duration of the tasks and the number of accesses to external documentation. We also collected qualitative data regarding the experiment format and the tools used to identify possible effects on the outcome of the experiment, with the help of an online form. Together, these elements help us answer the research questions and support the discussion of the validity of the hypothesis. "Empirical Study" describes in detail the experimental design, data analysis, and limitations of our study.

## RELATED WORK

Information about the pattern instances can be made available by creating documentation describing such instances or by detecting pattern instances in the source code. In this section, we will explore some good documentation practices and the approaches or tools developed so far to tackle these problems.

## Good documentation practices

The works are analyzed in light of the patterns for creating consistent software documentation that have before been distilled by *Correia et al. (2009)* from practices and tools: *Information Proximity*, *Co-Evolution*, *Domain-Structured Information* and *Integrated Environment*. As far as we know, these patterns constitute the only collection that tries to capture recurring good solutions for keeping software documentation consistent. *Correia et al. (2009)* demonstrate the validity of these patterns through a few real-world known uses, as commonly done in the patterns community, and we highlight some of such known uses in the next paragraphs.

**Information proximity** addresses preserving documentation consistency when related information is scattered across documents/artifacts. Easing the access to artifacts that contain related information, by using *links* between artifacts, using a *single source*, *transclusion*, or *views*. This pattern has been used successfully in tools such as Javadoc

(*Friendly, 1996*), WikiWikiWeb (*Cunningham, 1995*), and others described by *Bayer & Muthig (2006)*.

**Co-evolution** addresses updating related information scattered across artifacts, more precisely, defining when it should be updated. It distinguishes between *synchronous co-evolution* (update immediately after changes occur) and *time-shifted co-evolution* (track pending changes and update them when needed). This pattern has been successfully used in tools such as Javadoc, WikiWikiWeb (*Cunningham, 1995*) and in the work of *D'Hondt et al. (2002)*.

**Domain-structured information** tries to solve the problem of structuring the information in documentation, to automatically assess consistency among the related artifacts. This is achieved by formalizing contents according to their domain concepts, avoiding multiple interpretations of the same information. This pattern has been successfully applied on DART (*Radev et al., 2020*) and other documentation tools such as *OpenAPI (2018)*.

**Integrated environments** try to solve the problem of maintaining consistency among independent but content-related artifacts. These environments allow handling several types of artifacts uniformly and provide a structure for them to interoperate with each other. This pattern has been successfully used by the Eclipse IDE, Visual Studio, IntelliJ IDEA, and other IDEs.

## Documenting pattern instances

Pattern instances have been documented through **text-based** approaches, such as code annotations or HTML pages, **graphical representations** like UML diagrams or by filling out **pattern code templates**.

### Text-based

This is the most commonly used approach. It can be achieved by annotating the source code with comments or by providing external documentation pages (*Sametinger & Riebisch, 2002*; *Torchiano, 2002*; *Hedin, 1998*; *Odenthal & Quibeldey-Cirkel, 1997*; *Hallum, 2002*). A good example of a tool that belongs to this category is Javadoc (*Li et al., 2007*). These approaches normally start by defining new tags for documenting pattern instances in different formats (Fig. 1). Even though most of these approaches assure information proximity, co-evolution is hardly supported. For example, *Sametinger & Riebisch (2002)* use links to automatically update the documents when the code annotations change. However, these annotations are not updated when the source code is modified, which may lead to inconsistencies.

### Graphical representation

Design patterns can be represented by standard notation UML diagrams, making this representation familiar to developers. As a result, some approaches believe that the ability to visualize the system's design patterns graphically, such as UML diagrams, can ease the comprehension of the software (*Tøese & Tilley, 2007*; *Schauer & Keller, 1998*; *Dong, Yang & Zhang, 2007*). However, if the system comprises many classes, this representation can become hard to understand (Fig. 2).

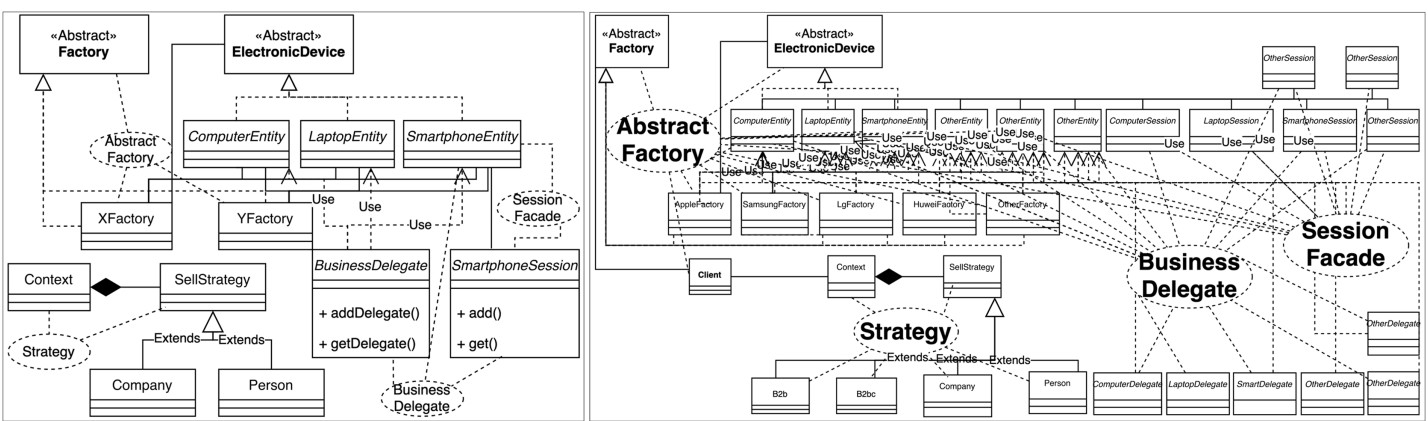

**Figure 1** A class with patterns documented (left) and an external documentation example of the same class (right), adapted from *Hallum (2002)*.

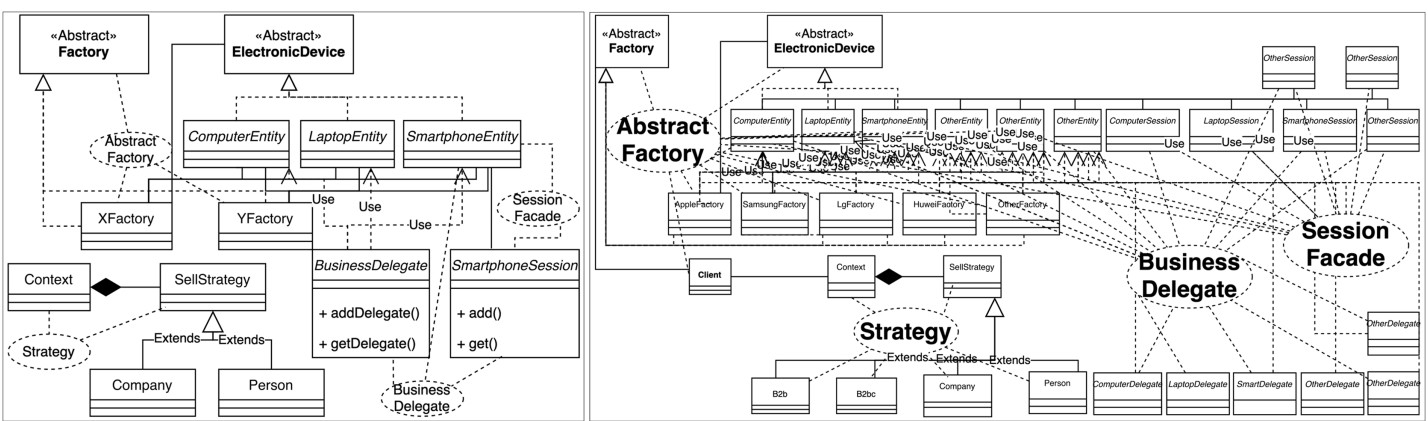

**Figure 2** A simple design pattern instance diagram using the standard UML notation (left) and a more complex diagram using the same notation (right), inspired by *Schauer & Keller (1998)*.

### Pattern code templates

Some approaches require developers to document design pattern instances by filling out pattern code templates. These approaches use code checkers, which validate specific pattern instances if they respect their specification, such as intent, rules, and participants.

**Table 1 Overview of approaches for documenting pattern instances.**

| Strategy | Approach | IP | CE | DSI | Validation | Liveness | Visualization |
|---|---|---|---|---|---|---|---|
| Text-Based | *Sametinger & Riebisch (2002)* | Yes | Yes | Yes | No | 1 | Code annotations, html pages |
| | *Torchiano (2002)* | Yes | Yes | Yes | No | 1 | Code annotations, html pages |
| | *Hallum (2002)* | Yes | Yes | Yes | No | 1 | Code annotations, html pages |
| | *Odenthal & Quibeldey-Cirkel (1997)* | Yes | Yes | No | No | 1 | Code annotations |
| Graphical | *Tøese & Tilley (2007)* | No | No | No | No | 1 | UML |
| | *Schauer & Keller (1998)* | No | No | No | No | 1 | UML |
| | *Dong, Yang & Zhang (2007)* | No | No | No | No | 1 | Javadoc html pages |
| Template | *Florijn, Meijers & van Winsen (1997)* | No | No | No | Yes | n/a | UML |
| | *Lovatt, Sloane & Verity (2005)* | No | No | No | Yes | 3 | n/a |
| | *Cornils & Hedin (2000)* | Yes | Yes | No | Yes | 3 | Boxes next to editor |

**Note:**
IP, Indicates which of the approaches support *Information Proximity*; CE, Indicates which of the approaches support *Co-Evolution*; DSI, Indicates which of the approaches support *Domain-Structured Information*; n/a, Information is not available.

## Discussion

We characterize the approaches for documenting pattern instances according to the good documentation practices introduced in "Good Documentation Practices" (information proximity, co-evolution, and domain-structured information), their liveness level, how they present the pattern instances descriptions and, whether they can validate the implemented patterns or not.

As we can see from Table 1, none of the presented approaches will simultaneously (i) maintain proximity between documentation and source code, (ii) update this documentation when changes to the source code are introduced and (iii) provide a liveness level above 3.

The closest approach to fulfill the three features mentioned above that we found was Cornils'. Still, it only warns developers that the code no longer respects the specified design pattern and does not support automatic co-evolution. Moreover, contacting the authors revealed that the tool is no longer available.

## Detecting pattern instances in code

Recovering design pattern instances from source code has been a topic of interest for the last two decades. This has led to the development of several approaches and tools to detect and visualize them. Some require the source code to be compilable; others also need it to be executed. Known as *design pattern detection tools*, they can be categorized in several ways, such as the level of automation, the level of liveness, the supported language, how they display the discovered patterns, and the type of analysis, among others.

These tools have been shown effective in recognizing pattern instances, but they have some limitations. Firstly, many do not scale when analyzing large systems, becoming significantly slower, especially those that perform dynamic analysis. Secondly, most of these tools are unavailable. Additionally, some are not up-to-date regarding the current programming language versions as they are no longer actively maintained. Finally, the results often contain many false-positive pattern instances, requiring manual confirmation

by the developer. The latter can be particularly challenging since developers inexperienced in design patterns may rely on the tool to guide them.

The design pattern detection tools were categorized according to the level of automation, the target's system language, how they display the detected pattern instances, if they are restricted to an Integrated Development Environment (IDE), their liveness level, and their limitations. To clarify, by *automation level* we refer to how independent they are, from the developer's interaction, which can range from manual to automatic. Table 2 overviews the 40 analyzed tools.

Through the analysis of the previous table, it is possible to see that two-thirds of the tools use a static analysis strategy. In other words, most tools do not need to run the target systems to detect design pattern instances. Nevertheless, there are some patterns that they cannot differentiate since they focus only on the static structure of the classes, ignoring their run-time behavior. An example is how most of these tools can not distinguish between the State and the Strategy pattern, which are structurally identical. Still, a few tools can do data flow and control flow analysis to infer behavior, such as **PINOT** (*Shi & Olsson, 2006*).

We can draw few conclusions regarding their automation level due to the difficulty of obtaining this information for many tools. A few of them require constant user interaction, throughout the detection process like **ePAD** (*De Lucia et al., 2010a*, *2015*) or **FUJABA** (*Niere et al., 2002*), and several that are completely autonomous, such as **PAT**, **DPAD**, **Columbus** (*Prechelt & Krämer, 1998*; *Zhang & Liu, 2013*; *Ferenc et al., 2002*). Note that, by *completely autonomous* we mean a maximum of one click, which is the execution of the tool on a given source code.

Most of these tools were designed to identify design pattern instances in C++ or Java. This might be because the design patterns they are trying to detect are those specified by the Gang of Four (GoF), who wrote the design patterns book in a C++ context (*Gamma et al., 1995*). Moreover, several of these design patterns described by the GoF are not useful outside this context or are defined significantly differently.

In terms of visualization, detected design pattern instances are displayed in several formats, including textual (plain or structured), Hypertext Markup Language (HTML), Extensible Markup Language (XML), or even in a graphical representation format like UML. We also noticed that most of the Eclipse plugins use a UML representation, which might be explained by this IDE's built-in ability to display diagrams in this format. Moreover, the tools that provide a graphic user interface like **SPOOL**, **MAISA** and **FUJABA** (*Keller et al., 1999*; *Nenonen & Gustafsson, 2000*; *Niere et al., 2002*), among others, also use UML representation for the detected pattern instances. As we have seen in "Graphical representation", the ability to visualize design patterns graphically can ease the comprehension of the software, which might be why these approaches tend to use this type of representation. Textual representations were mostly associated with command-line-like tools (*Diamantopoulos, Noutsos & Symeonidis, 2016*; *Shi & Olsson, 2006*).

Regarding IDEs, we have often encountered Eclipse. However, the amount of tools not restricted to any IDE is higher. Also, some tools can be used in different ways. For example,

**Table 2  Overview of approaches or tools to detect design pattern instances in source code.**

| Type | Tool | Authors | Automation | TL | V | IDE | Limitations |
|---|---|---|---|---|---|---|---|
| Static | PAT | *Prechelt & Krämer (1998)* | Automatic | C++ | Text | No | Unavailable |
| | SPOOL | *Keller et al. (1999)* | Automatic | C++ | UML | No | Unavailable |
| | HEDGEHOG | *Blewitt, Bundy & Stark (2001)* | Automatic | Java | Text | No | Unavailable |
| | SPQR | *Smith & Stotts (2003)* | Automatic | C++ | XML | No | Unavailable |
| | CrocoPat | *Beyer & Lewerentz (2003)* | Automatic | Java | n/a | No | Needs a pattern specification language |
| | Columbus | *Ferenc et al. (2005)* | Automatic | C++ | n/a | No | Unavailable |
| | PINOT | *Shi & Olsson (2006)* | Automatic | Java | Text | No | Can't detect pattern with incomplete data |
| | n/a | *Rasool & Mäder (2011)* | Automatic | Java, C# | Text, UML | No | Unavailable |
| | DPF | *Bernardi, Cimitile & Di Lucca (2013)* | Automatic | n/a | n/a | Eclipse | Needs a domain specific language syntax |
| | n/a | *Chihada et al. (2015)* | Automatic | n/a | n/a | No | Unavailable |
| | FINDER | *Dabain, Manzer & Tzerpos (2015)* | Automatic | Java | Text | No | Does not scale |
| | DP-CORE | *Diamantopoulos, Noutsos & Symeonidis (2016)* | Automatic | Java | Text | No | Detects few patterns by default |
| | DesPaD | *Oruc, Akal & Sever (2016)* | Automatic | Java | Text | No | Does not scale |
| | PatRoid | *Rimawi & Zein (2019)* | Automatic | Java | Text | No | Does not allow user feedback |
| | DPDML | *Oberhauser (2020)* | Automatic | Independent | Text | No | Incomplete |
| | GEML | *Barbudo et al. (2021)* | Automatic | Java | UML | No | Need to train the tool's algorithms |
| | PatternDetectorByDL | *Wang et al. (2022)* | Automatic | Java | UML | No | Web |
| | DPDF | *Nazar, Aleti & Zheng, 2022* | Automatic | Java | UML | No | Does not allow user feedback |
| | FUJABA | *Niere et al. (2002)* | Semi-auto. | Java | UML | n/a | Variants has to be defined explicitly |
| | n/a | *Tsantalis et al. (2006)* | Semi-auto. | Java | Text | No | Unavailable |
| | DeMIMA | *Guéhéneuc & Antoniol (2008)* | Semi-auto. | Java, C++ | Text, UML | No | Unavailable |
| | MAISA | *Nenonen & Gustafsson (2000)* | n/a | Prolog | UML, Text | No | Needs diagrams expressed as Prolog facts |
| | n/a | *Ferenc et al. (2002)* | n/a | C++ | UML, Text | No | Unavailable |
| | WoP | *Dietrich & Elgar (2007)* | n/a | Java | XML | Eclipse | Unavailable |
| | n/a | *De Lucia et al. (2007)* | n/a | Java | HTML | No | Unavailable |
| | D^3 | *Stencel & Wegrzynowicz (2008)* | n/a | Java | n/a | No | Unavailable |
| | n/a | *Rasool, Philippow & Mäder (2010)* | n/a | Any EA sup. lang. | n/a | VS | Unavailable |
| | MARPLE | *Arcelli Fontana & Zanoni (2011)* | n/a | Independent | UML | Eclipse | Unavailable |
| | n/a | *Mayvan & Rasoolzadegan (2017)* | n/a | Independent | UML | Eclipse | Unavailable |
| Dynamic | n/a | *Heuzeroth et al. (2003)* | Automatic | Java | n/a | No | Unavailable |
| | n/a | *De Lucia et al. (2009)* | Automatic | Java | UML | Eclipse | Unavailable |
| | n/a | *De Lucia et al. (2010b)* | Automatic | Java | n/a | n/a | Unavailable |
| | DPAD | *Zhang & Liu (2013)* | Automatic | Java | Text | Eclipse | Unavailable |
| | n/a | *Wendehals (2004)* | Semi-auto. | Independent | n/a | No | Unavailable |
| | DPVK | *Wang & Tzerpos (2005)* | Semi-auto. | Eiffel | Text | Eclipse | Unavailable |
| | ePAD | *De Lucia et al. (2010a)* | Manual | Java | UML | Eclipse | Low precision for some patterns |
| | ePADevo | *De Lucia et al. (2015)* | Manual | Java | UML | Eclipse | Unavailable |
| | PTIDEJ | *Guéhéneuc (2005)* | n/a | AOL, Java, C++ | UML | No | Users can't specify layout information |
| | n/a | *Li et al. (2007)* | n/a | C++ | UML | No | Unavailable |
| | n/a | *Lee, Youn & Lee (2008)* | n/a | Java | n/a | n/a | Unavailable |

**Note:**

**Type**, Indicates what type of code analysis *Technique* is used by the approaches; **TL**, Indicates the Target Language; **V**, Indicates how the *Visualization* of pattern instances is achieved by the approaches; **IDE**, Indicates which of the approaches are restricted to an *IDE*; **VS**, Stands for Visual Studio. NET; **n/a**. Information is not available.

**PTIDEJ** (*Guéhéneuc, 2005*) can be imported as a library for a Java, C++, or AOL project or used as a plugin for Eclipse.

Since feedback is only provided through a direct request by the user, these tools support the second liveness level. In fact, design pattern instances are only displayed after we explicitly execute the tools on a particular software system.

Additionally, we would like to highlight one interesting feature not directly contemplated in Table 2. Not all tools search the software systems for every design pattern they may find. Some, such as **PAT** or **DP-CORE** (*Prechelt & Krämer, 1998*; *Diamantopoulos, Noutsos & Symeonidis, 2016*) allow checking the source code for a specific design pattern. This could be interesting if, after the developer specified the design pattern he was implementing, the tool could constantly check if that design pattern is found. For this purpose, the pattern detection tool would be running in the background. A negative result could be used to inform the developer that the design pattern was not properly implemented.

Finally, we would like to highlight that many of the tools presented in Table 2 were proposed more than 10 years ago, which could give the impression that this topic has stopped progressing, been abandoned, or been considered resolved. We believe the prevalence of relatively old tools in our review is due to our focus on tools supporting the GoF design patterns, which were published in 1995 and were very popular in the second half of the 1990's and early 2000's. Therefore, many works proposing tools that support these patterns are also from this time. Since then, new design patterns continued to emerge, and we believe the approach presented in this article can be easily extended to support such new design patterns. Additionally, with the increasing popularity of artificial intelligence methods, we realize that new work aimed at detecting design patterns has taken advantage of these techniques mainly from 2019 onward.

## THE *DESIGNPATTERNDOC* PLUGIN

We developed a plugin for IntelliJ IDEA, named *DesignPatternDoc* to make it easier to switch-contexts between creating and documenting software, as detailed below.

### Live pattern instance documentation

Our approach uses liveness to tackle the problem of switching between the activity of programming and documenting. We focus on increasing the amount of feedback about the pattern instances in the source code and provide this kind of feedback anytime the reference to a pattern participant is inspected or edited by a developer. This intends to allow proximity between the artifacts and make some tasks easier to perform, like editing and creating documentation or comprehending the design of the software systems. Ultimately, since the increase in the liveness level makes it easier to maintain and consume documentation, it could also reduce inconsistencies between the artifacts. During the rest of the section, we explore the principles used as guidelines for the design of the approach, the prototype's implementation and architecture, and the features that cover that architecture.

The goal of this work is to address the challenge of **understanding** a software system in terms of its design patterns, by reducing the mutual feedback loop between programming and documenting, easing the transition between these phases. By doing this, we will also ease the **creation**, **consistency maintenance**, and **use** of pattern-based software documentation. We seek to design a tool with a high level of liveness that supports good documentation practices and provides visual awareness of the pattern instance descriptions and the program state to the developer (like displaying missing pattern participants). More specifically, our tool design has the supporting guiding principles described below.

- **Liveness** at a higher level can provide almost instant awareness about relevant aspects of a system. Higher liveness can mean instant access to the documentation of pattern instances when editing the source code, alerting the developer when the documentation is incomplete, and easier awareness of the effects of editing the documentation, among other benefits.

- **Information proximity** is a convenient strategy to access live feedback, if we look at the current state of source code and of documentation as possible forms of feedback of each other. More specifically, we can use links and transclusion to ease access to the pattern instances documentation directly from the source code.

- **Co-evolution** can be supported by liveness, as immediate feedback about possible inconsistencies between artifacts supports synchronous co-evolution. In other words, it enables to update of all the related artifacts every time a developer introduces a change. For example, after renaming a class that plays a role in a pattern instance, its documentation also needs updating. Doing so helps to avoid obsolete documentation.

- **Domain-structured information** allows feedback to be provided at a granular level. More specifically, this means structuring the pattern instances according to a data model, capable of representing them in a rich format, instead of using plain text to represent them.

- **Integrated environments** makes it easier to increase the level of liveness of one artifact in terms of another by maintaining all related artifacts (*e.g.*, source code and its documentation) under the same environment, reducing the need to switch context constantly when alternating between software development phases.

## Design and functionality of the plugin

We developed a prototype of a tool that follows the principles described above. This tool is a plugin for the IntelliJ IDEA Integrated Development Environment, which can analyze source code, live-suggest and generate pattern-based documentation for specific pattern instances, and display inconsistencies. Based on our principles, we have created a more concrete desideratum for the design of the prototype and illustrate with functionalities of the plugin whenever possible[1].

---

[1] Additional illustrations of the plugin capabilities can be found in the experiment materials, in particular in the instructions for the experimental group. See file `questionnaire/experimentalGroup.pdf` of the experimental package, which is accessible through DOI 10.5281/zenodo.10849701 (*Lemos & Correia, 2024*). The materials are described in more detail in "Empirical Study".

```
public class AnimalFactory extends SpeciesFactory {

    @Override
    public Animal getAnimal(String type) {
        if("Elephant".equalsIgnoreCase(type)){
            return new Elephant();
        }
        else if("Lion".equals
            return new Lion()
        }
        else {
            return new Pig();
        }
    }
}
```

This class may play the role(s) Product of the Abstract Factory Design Pattern.        ⋮

Add pattern instance documentation   Alt+Shift+Enter        More actions…   Alt+Enter

animals
public class Elephant
extends Animal                                                                         ⋮

**Figure 3** **Example of a pattern instance suggestion.**   

## Functionalities

### Identifying pattern instances in the source code

The plugin runs a detection tool a few seconds after the developer stops typing. This detection tool scans the project folder for the design patterns supported by it. This service is executed periodically and, in each execution, exports a set with all the design patterns (name, roles, participants) found. The output may be an empty set. The contents of this set are used to suggest pattern instance documentation to the user, as shown below. The plugin contains three different types of code inspections: one is responsible for highlighting the objects that we believe to play a role in a pattern instance; the second one is responsible for updating the persisted pattern instances after renaming an object; the third and last one is responsible for alerting the user when the source code is incomplete or inconsistent regarding its documentation (*e.g.*, missing a role).

### Suggesting detected pattern instances to the developer

The developer should be able to decide whether to accept or reject those suggestions. This is important since we do not want to be intrusive. The liveness in this context should be at level four. If there are any pattern instance suggestions, they should be instantly available to the user. Figure 3 shows the names of the classes that play a role in a pattern highlighted with a different background color, and a popup that is displayed when hovering one of those names with the mouse pointer and that features more information about the pattern instance in question.

### Renaming inspection

Instead of turning the documentation obsolete after renaming a certain object, it makes sure the documentation is kept updated. This code inspection supports the undo operation after renaming an object.

**Figure 4** Example of an incomplete documentation warning.

**Figure 5** Example of a pattern instance visualization.

### Incomplete documentation inspection

This feature highlights all the pattern participants from a pattern instance, where at least one role is not played by any object. Figure 4 illustrates this situation. This helps the user spotting which objects should be created to complete the design pattern. If the missing pattern participants were already implemented, it warns the developer that the documentation hasn't been updated yet. This highlighting provides two types of quick fix: (1) Edit the documentation or (2) Delete the pattern instance.

### Creating and displaying documentation in the IDE

This should reduce the context switching between software creation and documentation since both artifacts will be created and viewed in the same environment. Here, documentation visualization should correspond to, at least, the fourth liveness level. Figure 5 shows how the names of classes that participate in pattern instances are postceded

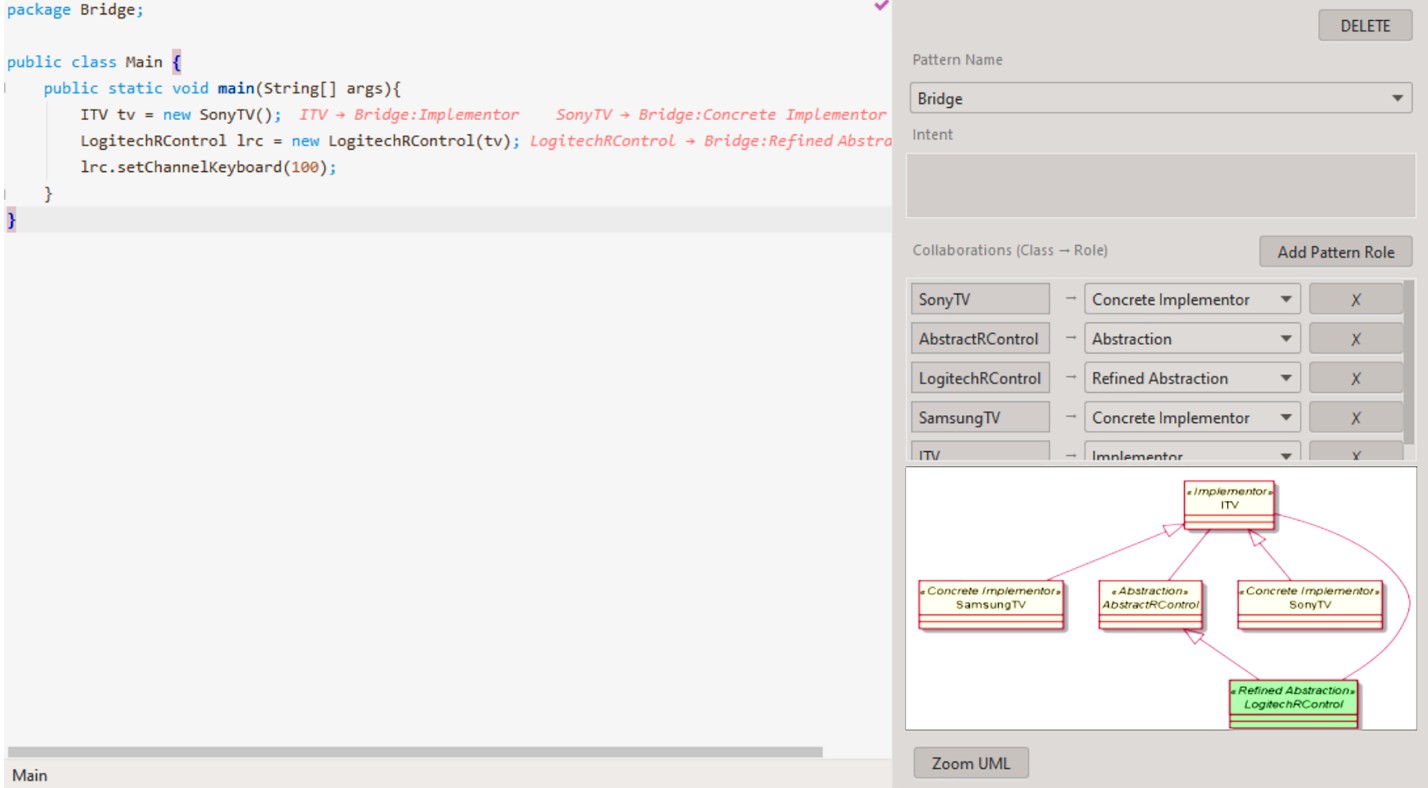

**Figure 6 Example of the live pattern instance editor.**

in the editor by a reference to the pattern and the role that the class plays in it. Hovering the class name with the mouse pointer also shows additional information about the pattern instance.

### Pattern instance live editor

Persisted pattern instances can be edited *via* the pattern instance live editor as presented in Fig. 6, located on the right side of IntelliJ's file editor. The content on this window is displayed if the mouse's cursor is on top of a class, that plays a role in at least one persisted pattern instance. Changing the cursor's position among the different class names, in each file, updates the window's content, accordingly. Any change to a pattern participant (name, role), will update the UML preview after a few seconds. Since large images have to be resized to fit the dialog, we provide a "Zoom UML" button (bottom of Fig. 6), which can be accessed to display the real-sized UML in a new window.

### Pattern hints

Pattern hints are text extensions that are appended in front of the lines where the class names of pattern participants are found (see Fig. 7). These extensions provide details on the roles played by that object in each pattern instance. The style and color were selected to mimic the text extensions displayed by IntelliJ debugging mode.

```
package abstractFactory;

public class Cat extends Animal {      Cat → Abstract Factory:Product   Animal → Abstract Factory:Abstract Product
    @Override
    public String makeSound() {
        return "meoww";
    }
}
}
```

**Figure 7  Example of pattern hints.**                

**Figure 8  Example of manual documentation.**         

*Manual pattern instance documentation*

Even though pattern instances can be persisted by accepting the tool's suggestion, manual documentation of a pattern instance is also possible. By right-clicking on a class name, the developer will be presented with that option (see Fig. 8). This will automatically place that class as one of the pattern participants of a default design pattern. At the same time, the pattern instance live editor will display the new pattern instance for further editing.

## Architecture

An overview of our prototype's conceptual architecture is depicted in Fig. 9. Two types of data store are represented: (1) the *Accepted Pattern Instances Store* and (2) the *Detected Pattern Instances Store*. The first one is responsible for persisting the pattern instances' documentation that was either manually created or the outcome of accepting a pattern instance suggestion. To store this information the plugin relies on IntelliJ IDEA to persist the plugin state as XML, which results in a `pattern_instances.xml` file. This file can be committed to the project's version control for an easy way to share it with other team members. The second data store is responsible for managing a collection of all the pattern instances detected by the selected tool (view the first desideratum in "Functionalities"). It is used as input for the component described in the second desideratum—*the Pattern Instance Suggester*. Having these as separate data stores allows different strategies, such as using persistent storage for one and volatile storage for the other. Namely, there is no need to persist the pattern instances found by the detection component, as it will scan the source code every few seconds. Since there will be potentially simultaneous read/write requests from multiple sources to the *Accepted Pattern Instances Store* component, it is also important to note that it needs to support concurrent access.

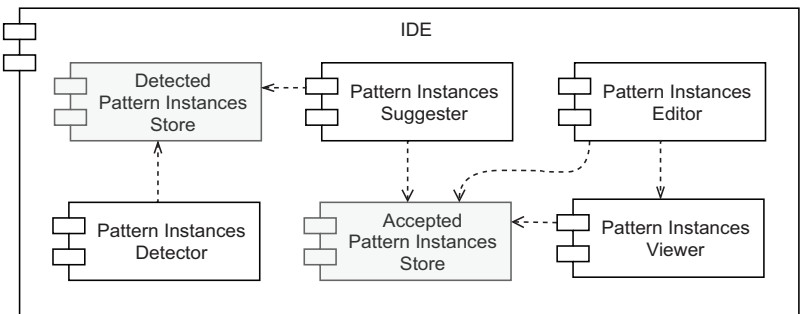

**Figure 9** **Prototype's conceptual overall architecture.**

As to the other components, the *Pattern Instances Detector* runs in the background every few seconds or a couple of seconds after the developer stops modifying the source code. Its output is stored in volatile memory. We designed the pattern detection strategy to be easy to replace, as long as the output matches the same format, using the Factory Method design pattern. The *Pattern Instances Suggester* component visually informs the developer every time a pattern instance is detected. The feedback that it provides is instantaneous and the least intrusive as possible. Moreover, it does not modify the persisted documentation without the explicit agreement of the developer, who can ignore the suggestions. On the other hand, accepting the suggestion will automatically generate the pattern instance documentation for that detected design pattern.

The last desideratum is attended by the *Pattern Instances Viewer* and *Pattern Instances Editor* components. As we want to reduce the feedback loop between programming and documenting, the documentation should be kept in proximity to the source code, avoiding the developer changing context to consume it and allowing easier detection of when they are not in sync. Given this, we designed a component responsible for providing the persisted documentation in the IDE, where the source code is edited. Namely, we allow the developer to inspect the reference to a given class, to view the pattern instances where it plays at least one role. This is the responsibility of the *Pattern Instances Viewer*. We defined how these pattern instances are visually represented using our findings in the literature review presented in "Related Work" and will describe it in more detail in the next section. The *Pattern Instances Editor* component allows editing persisted pattern instances and manually creating new ones in a *live* manner. The pattern instance viewer will immediately show any changes a developer applies to the documentation of a given pattern instance. Hence its direct dependency on the *Pattern Instances Viewer* component.

Finally, since we want all these components to co-exist in the same context, they should all belong to the same development environment. Given that source code is nowadays, to a large extent, created in IDEs, and we want documentation to be created and consumed in the same environment, we assume all of these components exist within an IDE.

## Design decisions

### Java language

Most of the existing approaches to detect design pattern instances are designed for Java systems (*cf.* "Related Work"). Many of these approaches are theoretical, or their implementation is not public—most articles do not reference a tool, nor were we able to find them by other means. We also filtered out the tools made for a specific IDE since reusing these would not be an easy task. This constrained our options regarding which programming language to support, as we sought to leverage one of the existing pattern instance detection tools rather than build one into our implementation, leading us to choose Java.

### Static analysis

Another important decision was regarding the source code analysis technique used by the detection tool. We decided to go for a static analysis tool. Since these tools do not require running the code being developed, they grant less computational effort for providing live feedback. Notwithstanding, there are some drawbacks, as discussed in "Detecting Pattern Instances in Code", but their overall performance is very satisfactory. From a hardware perspective, our design decisions guarantee that if the machine can run IntelliJ IDEA it will be able to efficiently run the plugin and guarantee the proposed liveness level.

### Based on DP-CORE

We experimented with the tools that respected these criteria and provided public access to their source code. Unfortunately, we only managed to run the DP-CORE tool successfully. This tool supports detecting a limited number of design patterns[2], but its performance seemed good enough to test our hypothesis. Moreover, it allows easy expansion to new pattern definitions through a declarative language, which allows overcoming the limited number of supported patterns, if needed. This tool is wrapped by the *Pattern Instance Detector* component in our architecture.

### Graphical visualization

Two approaches are most prominent for pattern instance visualization (*cf.* "Documenting Pattern Instances"): *text based* and *graphical representations*. Since pattern instances are often represented with UML diagrams, the latter would seem the most appropriate approach. Still, we have also seen that most of the graphical approaches rely on external documentation to display the diagrams, which went against our desideratum. Nevertheless, we found that PlantUML[3] is well integrated into IntelliJ IDEA and allows visual representation of UML diagrams directly in the IDE environment, resolving this disadvantage. It is, therefore, used by the *Pattern Instance Viewer* component in our design.

### Integrated into intelliJ IDEA

Lastly, we chose the *IntelliJ IDEA* IDE as the environment into which we would integrate these components and functionality for its widespread use (*Vermeer, 2020*) and extensibility, which allowed us to implement these features as a plugin.

[2] DP-CORE supports six patterns, all from the *Design Patterns* book (*Gamma et al., 1995*): *Abstract Factory, Bridge, Builder, Command, Observer* and *Visitor*.

[3] PlantUML is a tool for generating different types of diagrams from a simple domain-specific language. More information is available at https://plantuml.com.

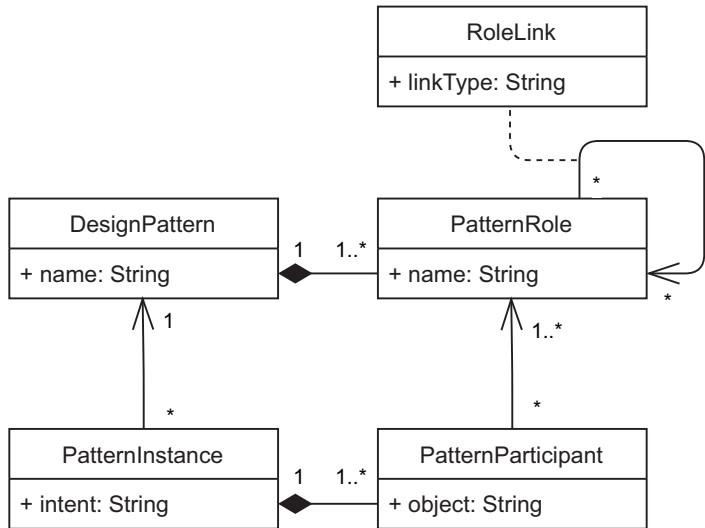

**Figure 10 Domain objects of the DesignPatternDoc plugin.**

### Domain model

Our prototype implements the domain model represented in Fig. 10. It represents *design patterns* and *pattern instances* using the TYPE OBJECT pattern (*Johnson & Woolf, 1996*), so that it is easy to extend the plugin with new patterns. The main goal of our approach is to allow documenting pattern instances of a software system. Each pattern instance is documented by describing the developer's intent when using that pattern and by identifying one or more pattern participants. Participants are the classes that play each respective role of the design pattern in the specific pattern instance. These classes are identified by their fully-qualified name in the *object* attribute (Fig. 11).

### Availability

The source code for the plugin is freely available in GitHub under the MIT License (https://github.com/SoftwareForHumans/DesignPatternDoc). Furthermore, it is published in IntelliJ IDEA's marketplace (https://plugins.jetbrains.com/plugin/14102-designpattern-doc).

## EMPIRICAL STUDY

Using a controlled experiment seems appropriate given our research goals. According to *Wohlin et al. (2012)*, "*Experiments are launched when we want control over the situation and want to manipulate behavior directly, precisely and systematically*". For this study, it is possible to control who uses one method (our plugin) and another (a simpler tool).

Therefore, we design and perform a controlled experiment to assess the approach described in "Live Pattern Instance Documentation", namely the effects of reducing the length of the feedback loop between *programming* and *documenting* activities, and more specifically to answer the research questions that we present at the beginning of this article

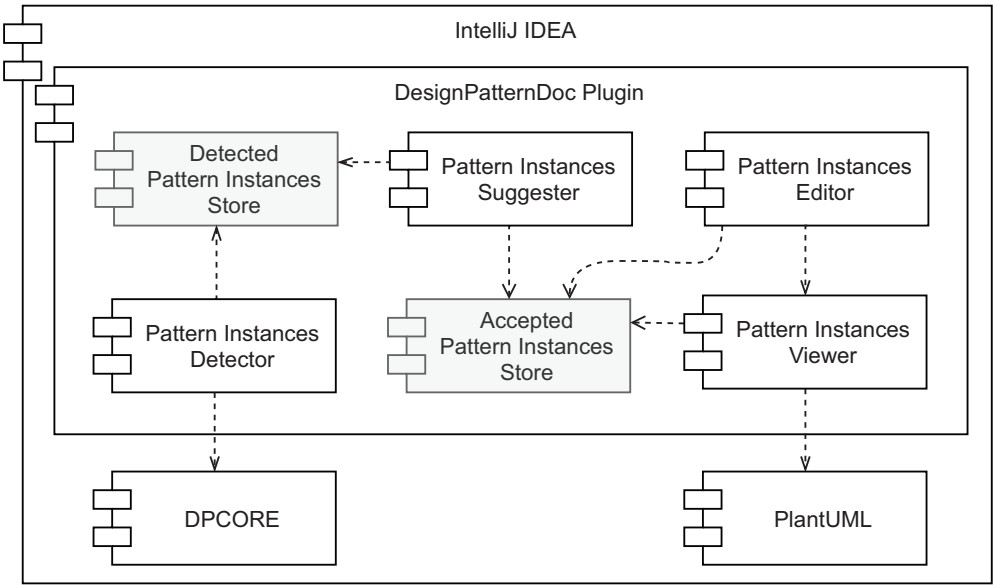

**Figure 11 Prototype's overall architecture.**

(*cf.* "Research Questions"). These research questions are answered with the help of specific dependent variables, as described below:

- **RQ1**: Investigates whether liveness would help improve the comprehension of the software's design. To help answer this research question we measure the **time** participants spend understanding and documenting the design of a software system.
- **RQ2**: Seeks to find out if it is possible to reduce the effort (time) to keep documentation updated by increasing the level of liveness. For this particular research question, we measure the **time** participants spend documenting a software system.
- **RQ3**: Evaluates if, by increasing liveness integrated into the IDE, we can observe a reduction in *context-switching*. To answer this research question we measure the **number of times** that the participant switches context between the IDE and the external documentation.

## Study design and conduction

This section details the different aspects considered when designing the empirical study. The materials we used, together with the data collected during the experiment, are available as an experimental package accessible through the DOI 10.5281/zenodo. 10849701 (*Lemos & Correia, 2024*). It includes the instructions and materials that were used by the participants (`questionnaire/*.pdf,TaskN/src/*` and `DesignPatternsReferenceCard.pdf`), the data collected from the questionnaire pertaining to task duration and code understanding (`answers/*Group.csv`), the data collected manually by the researchers pertaining to task duration and the number of context switches (`data.csv`), and the scripts for running the data analysis (`syntax.sps`). Due to a logistic error by the researchers, the source code and diagrams produced by the

participants were not preserved after the data analysis and, therefore, are not part of the replication package.

## Participants

Participants are admitted for participation in the study if they have previous experience in Design Patterns, and are randomly distributed among the *control* and the *experimental* groups. To ensure that this random assignment leads to an equivalent set of skills between the two groups of participants, some information about the participant's background is collected through a few questions. In our run of the experiment, we recruited the participants *via* an e-mail targeting students of two courses lectured at the Faculty of Engineering of the University of Porto—the Master in Informatics and Computing Engineering (MIEIC) and the Master in Software Engineering (MESW). In total, we had the collaboration of 21 students with 10 of them allocated to the control group and 11 to the experimental group.

## Data sources and variables

We collect data from five different sources: (a) the answers to assessment questions, (b) the source code produced in each task, (c) the pattern instances documentation produced in each task, in PNG format, (d) the tasks' duration, and (e) the number of times that the participant switches context between the IDE and the external documentation. The data collected from these sources provide the *dependent variables* for our study. We describe how they are measured in the *Data Collection* paragraphs, later in this section.

## Environment

The participants engage in the experiment remotely through a desktop connection, gaining access to a pre-configured development environment. In particular, participants are given access to IntelliJ IDEA, with or without the *DesignPatternDoc* plugin, depending on the group. Participants are given a different source code folder for each task, with a few classes in each folder, amounting to around 100 lines of Java code on average. For some tasks, the plugin has some pattern instances documentation, whereas, for the control group, we provide a PDF file with the same documentation. Additionally, each participant has access to a GoF design pattern reference sheet, available during the entire experiment. Both resources (PDF and reference sheet) make what, in this article, we often refer to as *external documentation*.

## Procedure

The participants are first assigned to the groups, corresponding to the two different treatments—the control group (CG) use IntelliJ IDEA, and the experimental group (EG) use IntelliJ IDEA with the *DesignPatternDoc* Plugin. The session itself takes around 50 min for each participant and is structured according to the following steps.

- **Background questions (1 min)**

Before starting the tasks, the participants are submitted to a small questionnaire to determine how comfortable they are with the tools, technologies, and concepts required for the experiment. This is used to discard the possibility of statistical deviations caused by skill dissimilarities between the two groups.

- **Plugin walkthrough—EG participants only (2 min)**

We want to avoid any time required to learn to use the *DesignPatternDoc* plugin can substantially affect the performance of the experimental group during the tasks (*cf.* the next step). To this end, we provide a quick overview of the tool to the participants in this group, to help them gain some familiarity with it, where a screenshot and a small description illustrate each plugin feature.

- **Tasks (45 min)**

The participants complete four programming and documentation tasks, which include (1) identifying the pattern instances present in a software system, (2) documenting a software system in terms of its pattern instances, (3) completing a system's implementation by exploring the provided pattern instance documentation and, finally, (4) the complete cycle —understanding, documenting and expanding a system.

- **Assessment questions (2 min)**

The participants answer questions designed to assess effects that are difficult to measure directly in an objective way, such as the extent to which the participants find the tool *useful*.

## Data collection

Much of the data is collected *via* a Google Form, including: (a) the total duration of each task, (b) the modified source code itself, (c) the produced pattern instance documentation and (d) the answers to the questionnaire items. The questionnaire is designed using five-level Likert items (*Likert, 1932*) with the format: (1) strongly disagree, (2) disagree, (3) neutral, (4) agree and (5) strongly agree. The complete form is part of the experimental package referred in the beginning of this section, and includes the following tasks:

T11. [The provided source code sample] contains one or more pattern instances. Which design pattern(s) are represented in the system?

T12. Document the pattern instances that you have found as a UML class diagram. Do it as was instructed previously, using [the tool], and specifying pattern roles as class stereotypes.

T21. John is trying to implement a simple system for controlling the light of a lightbulb, in his house. Unfortunately, he can't get it to work. Which pattern participant(s) are missing [in the provided source code sample]?

T22. Create new objects and/or modify those already provided to complete the system.

T31. Identify the main GoF pattern in this code and explain what changes (objects, pattern roles) would you need to apply to the system to contemplate [the provided set of] requirements.

T32. Implement those changes and add the pattern instances documentation required to understand the extended system. (Here submit only the [documentation])

Some data is collected manually during the experimental sessions by one of the researchers, who has screen access to the environment used by the participants. The researcher mainly acts as a spectator and measures the time of each task (including sub-tasks) using a stopwatch. During the execution of the tasks, he also counts the number of context switches—in other words, the number of times a participant leaves the IDE to access the external documentation (or the other way around). Specifically for the EG, he counts the times each participant uses the documentation features, such as viewing the generated pattern instances when hovering code elements with the mouse pointer or *via* the live editor. The participants are asked to submit their answers every time a task is completed, and to *think aloud* to help the researcher understand their intent and follow changes in context, like moving to the next question or accessing the external documentation. The researcher intervenes only at the end of each task to inform the participant whether the result is correct. This includes validating the source code and the documentation that participants are asked to produce and ensuring that the total time spent during a task allows for achieving working solutions.

## Data analysis

We perform the data analysis with the help of the SPSS Statistics software. The collected data—tasks' duration, number of accesses to internal or external documentation, and questionnaire answers—are aggregated in a single data file. This is included in the replication package in the `CSV` and `SAV` formats. The `SPS` file contains instructions in SPSS's command syntax language, and running it with the `SAV` file, produces descriptive statistics and results for hypothesis testing, using t-tests for variables with normally distributed data and Mann-Whitney U tests for the remaining variables.

## Pilot experiments

The experimental design is tested with pilot studies to identify unsuspected issues in the developed plugin, instructions, or data collection. The pilots follow the same protocol and environment meant to be used during the controlled experiment. They allow us to identify minor issues in the instructions that we fix for the *real* run of the experiment. In our run of the experiment, we conducted three pilots.

## Data analysis

Throughout this section, the control and experimental groups are, respectively, denoted by CG and EG. The *null* and *alternative* hypotheses are, respectively, denoted by $H_0$ and $H_1$. In each of the analyses below, we take $H_0$ as no difference existing between the two groups but $H_1$ will vary according to the specific analysis at hand. Other symbols that we use include: $n$, the sample size; $u$ and $r$, respectively, the u-statistic and the effect size of Mann-

Whitney U tests; $t$ and $d$ respectively, the t-statistic and the effect size of t-tests; $\rho$, the probability that $H_0$ is true; and $\alpha$, the significance level of statistical test results.

The experiment had the collaboration of 21 participants ($n$), which translates to 19 degrees of freedom when interpreting statistical tests. Moreover, we use a 5% significance level ($\alpha$) when interpreting the results of statistical tests and the guidelines proposed by *Cohen (1988)* for interpreting effect sizes.

### Sample background

We evaluate the participants' background through a questionnaire about their comfort with the concepts and technologies used in the experiment. This attempts to ensure that the two groups are similar and that any differences in the performance of the groups are a consequence of the tools provided during the experiment. The questionnaire results are in Table 3 and show no meaningful differences between the two groups.

### Task duration

We wanted to investigate if the time spent in the execution of the tasks by the control group was significantly greater than the time spent by the experimental group ($H_1 : CG > EG$).

We started by inspecting the total time spent by the participants on the execution of the programming tasks. The box plot presented in Fig. 12 shows that the CG indeed spent more time than the EG, but this high-level analysis is not very useful to answer our research questions (*cf.* "Research Questions"), so we also tested $H_1$ for *each* task.

Figure 13 allows to see a clear difference in the completion times for most tasks of the two treatments, which supports our hypothesis. Nevertheless, we also submitted the hypothesis to statistical tests, and the results can be seen in Table 4. Note that the time spent (seconds) during tasks T11, T22, and T32 does not follow a normal distribution; therefore, we resorted to MW-U tests for such tasks and t-tests for the remainder.

These results allow us to reject $H_0$ and show the CG as **significantly slower** than the EG ($\rho < 0.05$) with large effect sizes ($r > 0.5$, $d > 0.5$) for all tasks except T22. In T21 the participants can lean on the provided documentation to discover which design pattern is at stake and the extent to which it is partially implemented. These test results lead us to believe that the EG effectively used the plugin's feedback to its advantage during that task. Conversely, T22 consists merely of completing the implementation using the knowledge obtained in T21. The plugin does not yet provide any particular support for this, thus putting the CG and the EG on equal footing. This is a flaw in our experimental design. Namely, we should not be expecting the plugin, in its current version, to have a positive impact on this task's duration, since it does not yet use existing documentation to suggest how a developer could complete the instance of the pattern.

In sum, we can conclude that for those tasks that involve understanding which pattern instances were being implemented by the system or, even the act of documenting the system using pattern instances, the time spent by the control group was significantly greater than the time spent by the experimental group.

**Table 3 Summary of the answers to the background questions.**

|  | CG | | EG | |
|---|---|---|---|---|
|  | $\bar{x}$ | $\sigma$ | $\bar{x}$ | $\sigma$ |
| BG1 | 3.5 | 0.40 | 3.6 | 0.41 |
| BG2 | 3.7 | 0.34 | 3.6 | 0.36 |
| BG3 | 2.6 | 0.40 | 2.8 | 0.33 |
| BG4 | 2.7 | 0.42 | 2.8 | 0.33 |
| BG5 | 2.6 | 0.40 | 2.9 | 0.34 |

**Note:**
BG1, At this point I am comfortable working with IntelliJ.
BG2, At this point I am comfortable programming in Java.
BG3, At this point I know well the GoF design patterns.
BG4, At this point I recognize GoF design patterns in code.
BG5, At this point I can implement GoF design patterns.

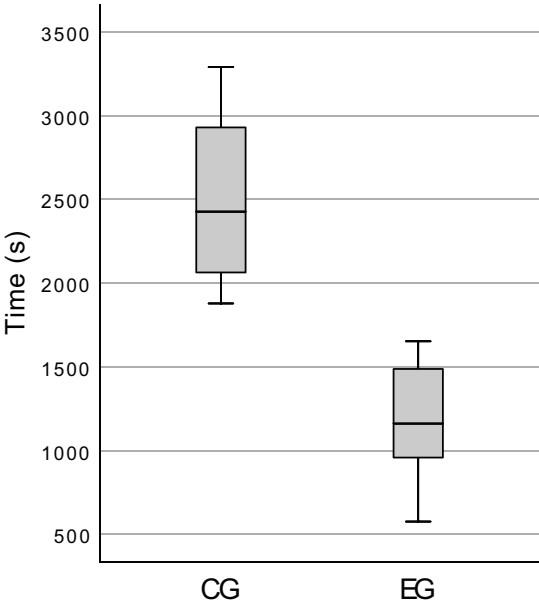

**Figure 12 Box plot of the total time spent by the participants, in the execution of the tasks.**

### Context switching

To evaluate if we can reduce the context switching between *programming* and *documenting*, we observed how each group accessed the external documentation. We believed that the context switching would be higher in the control group ($H_1 :$ CG > EG).

Running an independent-samples Mann-Whitney U test over our context-switching data produced the results seen in Table 5. We find that context switching is indeed **significantly greater** for the CG when compared with the EG ($\rho < 0.05$) with large effect sizes ($r > 0.5$) for every task except for T22. In the latter, we cannot reject $H_0$; we believe that the cause for this is the flaw in the experimental design that we have previously described (*cf.* "Task Duration"). Nevertheless, we find that, for most tasks, the participants using live documentation embedded in the IDE (*i.e.*, the EG) switched context significantly

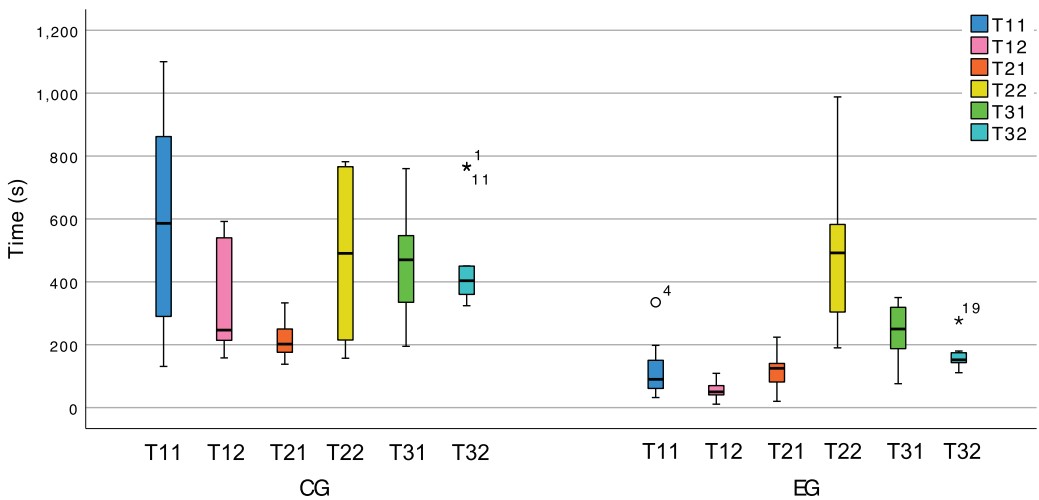

**Figure 13 Box plot of the tasks duration variable in both treatments.**

**Table 4 Summary of the statistical tests' results for the tasks duration variable.** The values of ρ are underlined when they represent statistically significant probabilities.

| | CG | | EG | | | MW-U | | | t-test | | |
|---|---|---|---|---|---|---|---|---|---|---|---|
| | $\bar{x}$ | $\sigma$ | $\bar{x}$ | $\sigma$ | $H_1$ | u | $\rho$ | r | t | $\rho$ | d |
| T11 | 585.8 | 104.66 | 117.3 | 26.80 | > | 6 | <0.001 | 0.57 | – | – | – |
| T12 | 325.5 | 52.10 | 55.55 | 8.44 | > | 0 | <0.001 | 0.71 | – | – | – |
| T21 | 211.8 | 18.70 | 114.9 | 17.24 | > | – | – | – | 3.816 | <0.001 | 1.67 |
| T22 | 475.3 | 80.41 | 483.9 | 70.57 | > | – | – | – | −0.081 | 0.468 | −0.04 |
| T31 | 468.5 | 51.43 | 243.1 | 26.64 | > | – | – | – | 4 | <0.001 | 1.75 |
| T32 | 462.2 | 52.41 | 163.3 | 12.93 | > | 0 | < 0.001 | 0.72 | – | – | – |

**Table 5 Summary of the MW-U statistic results for the context switching variable.** The values of ρ are underlined when they represent statistically significant probabilities.

| | CG | | EG | | | MW-U | | |
|---|---|---|---|---|---|---|---|---|
| | $\bar{x}$ | $\sigma$ | $\bar{x}$ | $\sigma$ | $H_1$ | u | $\rho$ | r |
| T11 | 4.6 | 0.792 | 0.45 | 0.207 | > | 1.5 | <0.001 | 0.71 |
| T12 | 4.9 | 1.016 | 0.18 | 0.122 | > | 2 | <0.001 | 0.73 |
| T21 | 3.3 | 0.367 | 1.09 | 0.251 | > | 6 | <0.001 | 0.6 |
| T22 | 1.5 | 0.563 | 1.91 | 0.530 | > | 46.5 | 0.289 | 0.02 |
| T31 | 4.3 | 0.895 | 0.91 | 0.285 | > | 11 | <0.001 | 0.48 |
| T32 | 5.5 | 0.543 | 0 | 0 | > | 0 | <0.001 | 0.84 |

fewer times, when compared with those with just access to external documentation (*i.e.*, the CG).

During the experimental sessions, the spectator collected data regarding the consumption of internal documentation (provided by the plugin). This was done by

counting accesses to (a) the documentation displayed when hovering the name of a class that participates in a pattern instance, (b) the pattern instance live editor, and (c) the plugin suggestions. Figure 14 shows the number of accesses to internal (ID) and external (ED) documentation during the software comprehension tasks (T11, T21, and T31). This puts into context the result of the statistical tests, given that a likely cause for different numbers of context switches is the availability of internal documentation within the IDE.

Regarding specifically the documentation creation tasks (T12 and T32), which required the participants to represent pattern instances as a UML class diagram, we found that the EG leaned almost exclusively on internal documentation. This is readily apparent in Fig. 15, and we believe that it can be attributed to the EG finding all the information that they required, within the IDE, as well as the means to perform this type of task.

### Assessment questions

To understand the participants' perception regarding the difficulty of the tasks, we asked them to answer three final questions. More specifically, (1) if it was easy to identify design patterns on the source code ($H_1$: CG > EG), (2) if it was easy to document the code using pattern instances in UML format ($H_1$: CG > EG), and (3) if the communication environment (remote computer) had a negative impact in their performance ($H_1$: CG $\neq$ EG). We start by plotting the data collected for the first two questions in Fig. 16, which shows that the EG found the identification and documentation tasks easier than the CG did.

We also run Mann-Whitney U tests to validate these hypotheses, and the results can be seen in Table 6. The answers to the first two questions allow us to reject $H_0$ and show the CG as having **significantly greater** values when compared with the EG ($\rho < 0.05$). This means that the participants agree that our approach does indeed help to improve software comprehension (medium effect, r > 0.3) and that it eases the documenting process (large effect, r > 0.5). Moreover, the answers to the last question do not allow us to reject $H_0$ and we find **no significant difference** between the two groups ($\rho > 0.05$). Even though nothing can be concluded from this test, we find that the large majority of participants do not believe that their performance was negatively affected by the remote environment (*cf.* Fig. 17).

### Discussion

The empirical study, as reported in the previous section, is designed to evaluate essentially these metrics: (a) **the time spent understanding** and (b) **documenting** a software system, and (c) **the number of context switches**, between the IDE and the external documentation. We rely on these metrics to answer our research questions (*cf.* "Research Questions"), with the goal of studying the consequences of increasing the liveness of software documentation based on pattern instances.

During tasks T11, T21, and T31, the participants are asked which pattern instances exist in the source code to evaluate the first metric. In "Task Duration" we report that the participants without access to our tool took significantly longer than those with access to it. Therefore, our approach seems indeed to reduce the time required to understand a

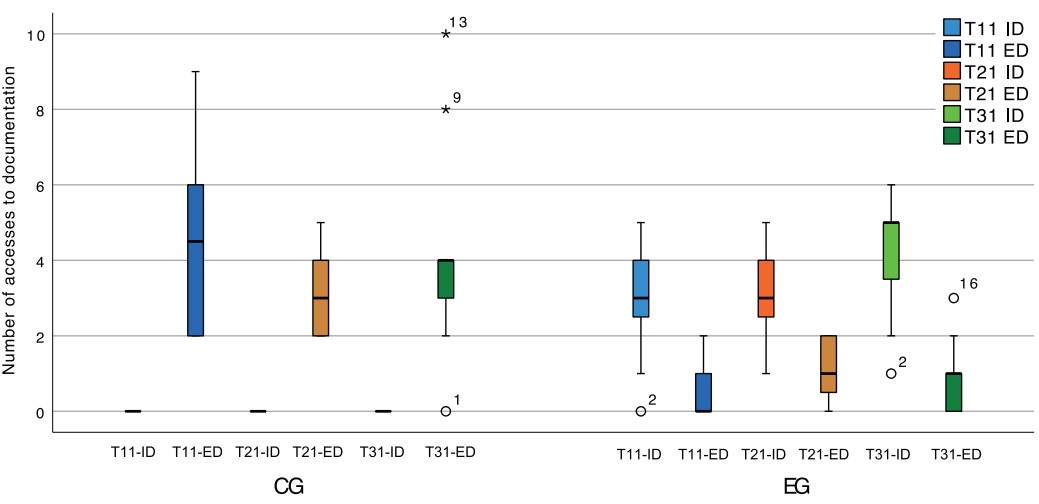

**Figure 14** Box plot of the number of accesses to internal (ID) and external (ED) documentation, in the software comprehension activities.

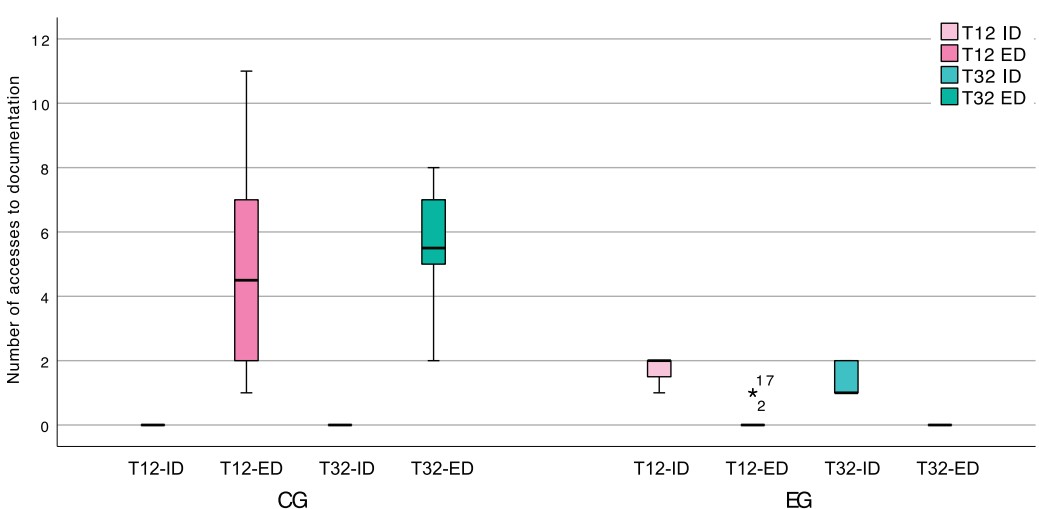

**Figure 15** Box plot of the number of accesses to internal (ID) and external (ED) documentation, in the software documentation activities.

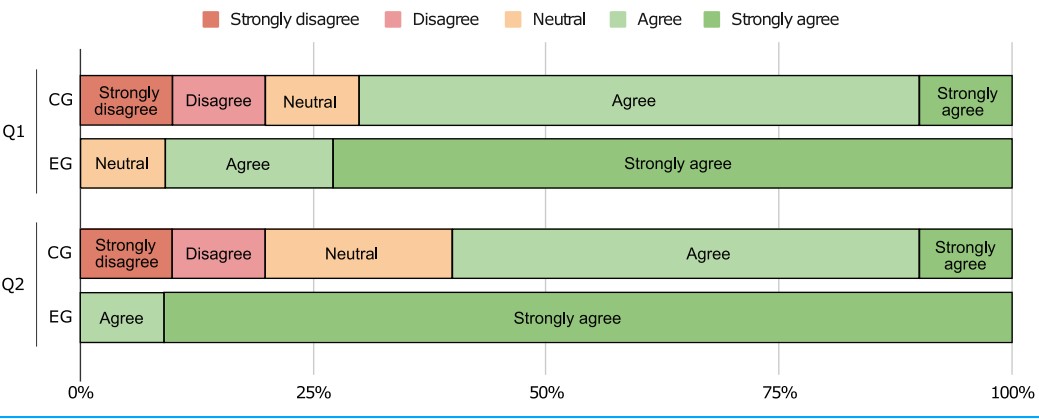

**Figure 16** Responses to final questions (1) and (2).

**Table 6 Summary of the MW-U statistic results for the final questionnaire.**

| | CG | | EG | | MW-U | | | |
|---|---|---|---|---|---|---|---|---|
| | $\bar{x}$ | $\sigma$ | $\bar{x}$ | $\sigma$ | $H_1$ | u | $\rho$ | r |
| Q1 | 3.500 | 0.373 | 4.640 | 0.203 | > | 19.500 | 0.003 | 0.34 |
| Q2 | 3.400 | 0.371 | 4.910 | 0.091 | > | 8.500 | 0.000 | 0.61 |
| Q3 | 1.300 | 0.213 | 1.820 | 0.325 | ≠ | 40.000 | 0.145 | 0.08 |

Note:
Q1. I found it easy to identify design patterns in the source code; Q2. I found it easy to document the code using pattern instances in UML format; Q3. The communication environment (remote computer) had a negative impact in the experiment.

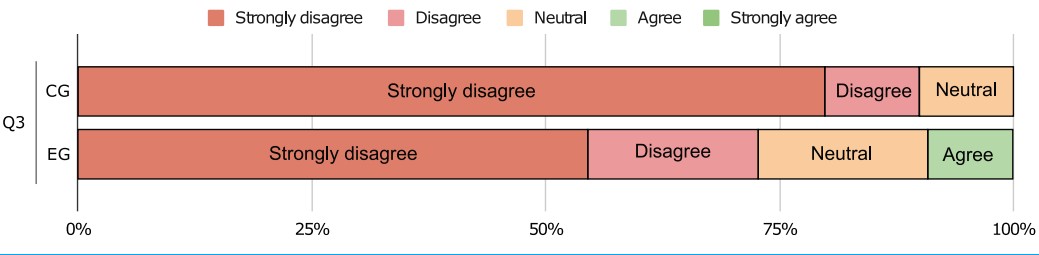

**Figure 17 Responses to the question if a remote environment had a negative impact in the experiment.**

software system. This result can be mapped directly to the first research question: *to what extent liveness can make it easier to understand a software system's design in terms of its pattern instances?*

Before answering this question it is important to reflect on these three tasks and why they should be evaluated together. It would be legitimate to wonder if the results for CG and EG would be equivalent in the scenario we would solely consider T11 to measure the time spent understanding the software, and if the CG used a design pattern detection (DPD) tool (see "Detecting Pattern Instances in Code") with support for the second liveness level. Therefore, we also used tasks T21 and T31 to help evaluate the first metric, since we expected these tasks to benefit from liveness, especially levels 4 and 5.

Our approach supports liveness levels 4 and 5 in particular through a few of the plugin's features. We believe the features *Identifying Pattern Instances*, *Incomplete Documentation Inspection* and *Pattern Hints* (see "Design and Functionality of the Plugin") are the main ones responsible for facilitating the understanding of the documentation. While the CG used 1266 TU/P[8] to perform these three tasks, the EG used 475 TU/P to perform the same tasks. This data shows that by using our approach EG completed tasks 62% faster compared to CG, therefore our approach positively influenced the comprehension of the software's design.

During tasks T12 and T32 the participants are asked to implement changes in the source code and document these changes, and we try to evaluate the second metric. In "Task Duration" we report that participants using our tool were significantly faster in these tasks than those from the control group. Therefore, we can say that our approach also reduced the time required to document a software system. This result can be mapped directly to the

[8] TU: Time Unit. P: participant.

second research question: *to what extent liveness can make it easier to keep software documentation updated?*

Although the results of tasks T12 and T32 can bring light to this question, it is important to mention that task T12 is similar to T11, as both can benefit from DPD tools. Knowing this, we should focus on the result of task T32 because it can benefit from approaches supporting liveness levels 4 and 5. For this task in particular, while the CG used 468 TU/P, the EG used 243 TU/P. Therefore, the EG completed this task 48% faster compared to CG. We credit this result in particular to the features: *Suggesting Detected Pattern*, *Incomplete Documentation Inspection*, *Pattern Instance Live Editor* and *Pattern Hints*.

It is also important to reflect that software documentation may easily become outdated when the act of documenting does not follow the evolution of the software. One of the reasons why this happens is because programming and development are not always performed together, making it easy for important information to get forgotten in the process. Additionally, since these activities are firmly related to each other, and involve different types of artifacts, transitioning between them implies a constant swap of context. Our approach addresses this concern by placing these different types of artifacts within the same environment, embedding documentation artifacts in the same place where the code is edited (IDE), and simplifying the act of documenting. Since the experiment showed evidence that, with our approach, developers reduce the time needed to document a software system, we believe that we provide the means to keep the documentation more easily updated.

Finally, we evaluated the number of context switches. In "Context Switching" we report that participants using the documentation provided by our tool within the IDE, have switched context significantly fewer times than those using only external documentation. Additionally, we noted that the participants were able to rely exclusively upon our tool to perform the documenting tasks. During the software comprehension tasks, the participants also reduced their need to use external documentation by inspecting the embedded documentation more often. This result can bring light to the third research question: *to what extent liveness can reduce context switching between programming and documenting?*

Our experiment showed that by increasing the level of liveness, by keeping implementation and documentation artifacts in the same environment and ensuring instant feedback, it was possible to reduce the number of context-switching by 89%. While the CG with 10 participants switched context 46 times, the EG with 11 participants switched context only 5 times.

## Threats to validity

During the design of the experiment, we considered any experimental conditions that could cause deviations in the results and, therefore, compromise the validity of our approach. Since we want to obtain sound answers to our research questions, we took precautions to discard these threats.

### Different skills

The design of the experiment assumes that the participants in the CG and the EG have similar skills and knowledge. Otherwise, we could not be certain that the differences in the results between the two groups were entirely due to the different treatments used and could perhaps be attributed to dissimilarities in the groups. Therefore, we selected participants with similar backgrounds (all MSc students from courses strong on software engineering topics) who had similar contact with design patterns. Additionally, we asked participants to answer a background questionnaire to statistically confirm that no significant differences existed between them.

### Internal factors

One of the goals of our experimental design is to ensure that both the CG and the EG are under the same conditions except specifically in what concerns the treatment—*i.e.*, the use of our liveness-focused approach to documenting pattern instances embodied by the *DesignPatternDoc* plugin.

Notwithstanding, the CG did not have access to any software tool for extracting pattern instances from source code while the EG did, as part of the plugin. Therefore, we could argue that the differences between the two groups go beyond being just able to benefit (or not) from liveness. In other words, we can ask ourselves if access to a pattern-instance extraction tool is playing a confounding factor in the association between using a live approach to documenting pattern instances and the duration of the tasks in our experiment. While we consider this a valid concern regarding the duration of software comprehension activities (T11, T21, and T31), we also believe that it is not likely that it influenced documenting activities (T12 and T32), for which a shorter duration in the tasks was also observed.

In any case, to fully address the concern, we can envision a different experiment: one where the CG also has access to a pattern-instance extraction tool, even if we expect that any gain in the duration of the tasks could be easily lost by the increase in context switching required by using another tool.

Furthermore, more factors could affect the results. To more reliably compare the two groups we asked the CG to produce manually drawn diagrams, close to those generated by our plugin. But manually drawing diagrams is qualitatively different from annotating pattern roles in source code, so the differences between the groups might not be only because of increased liveness. An alternative design could have the CG use text annotations instead of diagrams, but this option implies a relevant trade-off: it could allow for a more reliable assessment of the effects of liveness during *documenting* activities, at the expense of reliably assessing it during *programming* activities, given that CG and EG participants would be looking at different representations (respectively, text-based and diagrams) when trying to understand and evolve existing code.

To address this concern, we can envision future experiments with smaller scopes, focused specifically on *documenting* or on *programming* activities. While such experiments will not allow to evaluate our approach as a whole, they may allow to better isolate the role of liveness in observable effects.

Yet another internal factor to consider is whether the use of different terms in the task descriptions of the CG and EG may have biased the results. The goal of these differences was to avoid possible doubts of participants when following the instructions. Although we have no way of fully discarding this possibility, we are convinced they did not influence the results, as these small differences consist only of referring to different things that played the same role on each of the groups (*draw.io vs.* the *plugin*, and the *class diagram* drawn by the participant *vs.* the plugin-based *documentation* created by each participant).

### External factors

The option to use a remote environment had the benefit of making it logistically easier for students to participate in the study but opens a few threats to validity since the results could be affected by the communication environment, such as latency or network breakdowns. Moreover, it could make it harder for the researchers to observe the participants, which could be crucial for understanding the entire process toward a solution. The first issue was mitigated by submitting the participants to a final questionnaire, where they were asked if the communication environment affected their performance. For the latter, we had a spectator to watch the entire experiment by having access (*via* a remotely shared screen) to the computer where the experiment was taking place and by asking the participants to *think aloud*.

### Generalizability to professionals

Given that recruiting students is easier than recruiting professional software developers, we chose them as the subjects for evaluating our approach. This had other benefits, such as ensuring that every participant had a similar set of skills. However, we cannot help to wonder if the conclusions we take can be generalized to professionals due to differences in experience that will naturally exist. To tackle this concern, we will replicate this experiment with professionals in the future.

### Generalizability to different cultures

The participants in our study share the fact that they had all been recently exposed to the importance of design patterns and actively practiced using them in implementations and for communicating. However, we accept that not all teams will have design patterns ingrained in their culture and can see the value of creating documentation based on pattern instances. In such contexts, we do not expect our approach to bring benefits or even to be welcome.

## CONCLUSIONS

This work was conducted with the purpose of streamlining the process that comprises the *creation* and the *documentation* of software, making it easier to switch between the activity of programming and documenting, in particular, describing its design pattern instances. This objective lead us to three research questions presented in "Research Strategy".

Our hypothesis is that by increasing the level of liveness of documenting pattern instances, we will streamline the process of switching between programming and documenting, making it easier to understand a system's design and document it. Software

development and the creation of its documentation are often treated as different phases in the overall software development process. Developers have to constantly switch contexts, which most of the time, leads to loss of important knowledge, and inconsistencies among artifacts and, subsequently, has a critical impact on the learning process and reuse of software systems.

In order to evaluate this research hypothesis and answer the research questions we defined an approach based on the fundamental ideas of good documentation practices and live programming, implemented a prototype named DesignPatternDoc as a plugin for IntelliJ IDEA, and conducted a controlled experiment with MSc students of two courses lectured at the Faculty of Engineering of the University of Porto. As a result of the experiment, we found that the tool reduces the mutual feedback loop between the programming and documenting phases, helping the overall understanding of the software's design. This was achieved by adding live feedback to the documentation based on pattern instances. Our prototype provides feedback to the developer, while the system is being implemented, regarding which pattern instances should be documented. Furthermore, it generates the required documentation for a specific pattern instance, allows live editing of the persisted pattern instances, and alerts the developer when a pattern instance is incomplete (*i.e.*, when one of the pattern roles isn't being played by an object).

The answers to our research questions are discussed in "Discussion" based on the data collected in the experiment. We found clues that, with our proposed approach, (1) developers spend less time understanding a software system, (2) developers spend less time documenting a software system, which has the potential to make it easier to keep software documentation updated, and (3) the context switching between the IDE and the external documentation is reduced when embedding the latter in the IDE.

# FUTURE WORK

Our next steps in this research can be organized along three different aspects—(a) the approach, (b) the prototype, and (c) the empirical evaluation.

## Approach

### Increase the liveness level

This work was able to push liveness to level 4 according to *Tanimoto (2013)*'s six levels to define liveness, but we could consider increasing it further. While achieving level 6 seems still a considerable challenge, we can envision concrete use cases to realistically achieve level 5. For example, after defining that a class plays a role in a design pattern, the environment could suggest scaffolds for the classes that would play the rest of the roles for that design pattern. Further work will explore this and other level-5 features of the plugin.

## Prototype

### Expand the set of detected patterns

The set of supported design patterns that the plugin can detect in the source code would be interesting to expand. With DP-CORE this is easy to achieve for design patterns detectable

just by static analysis. We could also achieve this by replacing DP-CORE with an alternative detection tool that would cover a wider set of patterns.

### Pattern detection based on dynamic analysis

A way to widen the set of supported patterns would be through dynamic analysis of the source code. We want to explore if our approach can cope with such scenarios. With detection based on static analysis, we have managed to achieve level 4 of liveness; still, if the system needs to be *run* to detect the pattern instances effectively, it may reveal itself a challenge to maintain the quasi-real-time feedback that level 4 of liveness requires.

### Pattern instance view

In the same way that IntelliJ IDEA provides a *class view* and a *package view* for Java files, which allow browsing the sets of classes and packages in a project, the existence of a *pattern instance view* could provide easy access to all the pattern instances in a project. This view would allow navigating to a particular source code element from a specific pattern participant.

## Empirical evaluation

### Controlled experiment using more baselines

Our empirical study seeks to provide insights into how liveness can play a role in documentation based on pattern instances, and we chose external diagram-based documentation as a baseline for our comparison. However, comparing our approach with the most promising approaches and tools that we review in "Related Work" may lead to further insights. Unfortunately, most of the authors do not seem to provide a freely available tool that we can compare against, so doing such a study may prove unfeasible, depending on the specific approaches and tools we may want to compare with.

### Controlled experiment with professionals

Performing a controlled experiment with professional software developers could help us gain confidence in the generalizability of our study's results. To this end, the included experimental package can be, possibly with minor adjustments, applied to professional developers.

### Case study with professionals

Providing the prototype to a team developing a software product in an industrial setting could help to understand in-context the trade-offs that underlie the approach. It would support investigating how the approach behaves in a real-world scenario with a large number of classes and lines of code (*e.g.*, if the developers get overwhelmed by the number of pattern instance suggestions).

### Funding

This work was supported by FEUP and INESC TEC, and is co-financed by Component 5 - Capitalization and Business Innovation, integrated in the Resilience Dimension of the

Recovery and Resilience Plan within the scope of the Recovery and Resilience Mechanism (MRR) of the European Union (EU), framed in the Next Generation EU, for the period 2021–2026, within project HfPT, with reference 41. The funders had no role in study design, data collection and analysis, decision to publish, or preparation of the manuscript.

### Grant Disclosures
The following grant information was disclosed by the authors:
FEUP.
INESC TEC.
Resilience Dimension of the Recovery and Resilience Plan within the scope of the Recovery and Resilience Mechanism (MRR) of the European Union (EU).

### Competing Interests
The authors declare that they have no competing interests.

### Author Contributions

- Filipe Lemos conceived and designed the experiments, performed the experiments, analyzed the data, performed the computation work, prepared figures and/or tables, authored or reviewed drafts of the article, and approved the final draft.
- Filipe F. Correia conceived and designed the experiments, analyzed the data, prepared figures and/or tables, authored or reviewed drafts of the article, and approved the final draft.
- Ademar Aguiar conceived and designed the experiments, authored or reviewed drafts of the article, and approved the final draft.
- Paulo G. G. Queiroz analyzed the data, prepared figures and/or tables, authored or reviewed drafts of the article, and approved the final draft.

### Data Availability
   The data and materials used and produced in the context of our study are available at Zenodo: Lemos, F., & Correia, F. (2024). Experimental package for "Live Software Documentation of Design Pattern Instances" (v1.0) (Data set). Zenodo. https://doi.org/10.5281/zenodo.10849702.

### Supplemental Information
Supplemental information for this article can be found online at http://dx.doi.org/10.7717/peerj-cs.2090#supplemental-information.

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
