# Peer review of "Live software documentation of design pattern instances"

_PeerJ Computer Science, doi:10.7717/peerj-cs.2090_

## Round 0.1 · original submission · Minor Revisions

Dear authors,

Thank you for your resubmission to PeerJ Computer Science. The previous editor is not available but I was able to re-engage the previous reviewers to review the revised manuscript. Please note that the numbering does not match the original submission: reviewer 1 became number 3 and vice versa.

To my impression, the authors have revised the manuscript conscientiously and have addressed some of the weaknesses pointed out in the earlier review round. This can also be seen in the recommendations of reviewers 2 and 3, who are now in favor of accepting the manuscript.

The study is still subject to a number of limitations. However, the aim of a single experimental design and its reporting is not to overcome all potential limitations. That would be an impossible endeavor. Much more important is a complete reporting of the design decisions and discussion of identified limitations. The present manuscript does this well. Furthermore, the review of the available tools and the prototype evaluation are still valuable contributions.

Therefore, other than reviewer 1, I do not feel that a further major revision is required. However, I consider it necessary to polish the manuscript and its supplemental materials to meet the quality criteria of PeerJ CS.

In light of this, I kindly request the following minor revisions:

1) Begin the manuscript with a clear statement outlining the major limitations of the study to provide readers with a comprehensive understanding from the outset. For example, consider mentioning the sample size of the experiment and the scope of the prototype developed within the abstract, as suggested by reviewer 1.

2) In line with the journal's commitment to open access, it is recommended to make the supplemental materials (and possibly a release of the tool) permanently accessible to the public. Consider utilizing platforms such as Zenodo; individual releases of a GitHub repository can easily be mirrored on Zenodo. I also see that some of the data provided (e.g. answers/controlGroup.csv) contains links to google drive folders that can only be viewed with access rights. Here too, in the spirit of open access, making all relevant data permanently accessible to the public would be appropriate.

3) Figure 1 and Figure 2 are barely readable due to the low resolution. I recommend recreating the figures with higher resolution and using vector graphics or at least legible font sizes to better convey the message.

I encourage you to address these points in your revision, and I look forward to receiving the updated manuscript and supplemental materials. Should you require any further clarification or assistance, please do not hesitate to contact me.

Best regards,
Marvin Wyrich

**Language Note:** The review process has identified that the English language must be improved. PeerJ can provide language editing services - please contact us at [email protected] for pricing (be sure to provide your manuscript number and title). Alternatively, you should make your own arrangements to improve the language quality and provide details in your response letter. – PeerJ Staff

Reviewer 1 ·

Basic reporting

First, I would like to thank the authors for their rebuttal letter that details how they have addressed the concerns of the reviewers of a previous review round.

Regarding the basic reporting, one still finds:

(*) In the abstract and introduction, the study is not up front about the size and scope of the study. As a result, the reader has a hard time judging the contributions.

(*) The text still frequently mixes tenses, detracting from its overall clarity. This is particularly distracting when describing the experiment set-up in the past tense. It reads like a diary instead of prescriptive reproducible steps. In that respect, the changes have also introduced new issues.

(*) As far as I can tell, the raw data is still not supplied. It may have been in the previous round.

Experimental design

The paper is undoubtedly an improvement over the previous version. However, in my view the changes are not the major revisions the editor has asked for. Instead, the authors elaborate and rephrase the paper in relatively minor changes that leave the paper and its structure largely intact.

Key concerns of the reviewers and the editor regarding the methodology have not yet been sufficiently addressed. Most importantly, the experimental design is still lacking with respect to the recommendations of the editor.

Validity of the findings

Many of the comments of the previous reviewers still apply.

Cite this review as

Reviewer 2 ·

Basic reporting

My summary basically remains the same: The paper proposes "live documentation" as a part of live software development and evaluates it as a controlled experiment. It argues that it is better to keep the process of programming and documentation together in one environment. It looks at related work and presents and implements the approach where design patterns can be detected and annotated to code and visualized live and directly in the IDE. To evaluate this in a controlled experiment, a control group (CG) has to identify and document design patterns in various tasks. It is measured how long they take and how often they have to switch between tools. The experimental group (EG) has to perform similar tasks but uses an enhanced development environment that allows the documentation and visualization of design patterns directly in the programming environment. The experiment shows that the EG is significantly faster. The authors added

The authors improved the paper in various places to address the issues we reviewers had. They improved and extended the description of functionality, which helped a lot in getting a better picture of how the proposed product works. They added a new discussion section to address the various issues we reviewers had. The new conclusion section is an improvement.

My critique points I had in the original review are now all addressed:

- There is now a better relation to Tanimoto's (2013) "six levels to define liveness"
- The description of the actual approach is very good now
- It is not clear how patterns are persisted

Experimental design

Regarding the experimental design, I had the following issues:

Critique:
- The tasks the users had to perform are not explicitly stated in the paper, but they are only paraphrased and it is not clear what the participants had to do in for example in T12. After looking into the material, one can assume that Task 1.3 is meant.
- The tasks the users had to perform are sometimes different in the CG and EC. The qualitative difference is not discussed. E.g.
- T12, Task 1.3. in CG: "Document the pattern instances that you have found as a UML class diagram. Do it as was instructed previously, using draw.io, and specifying pattern roles as class stereotypes"
- T112, Task 1.3. in EG: "Document the pattern instances that you have found as a UML class diagram. Do it as was instructed previously, using the plugin, and specifying pattern roles as class stereotypes."
- The paper subsumes the differences as not in the same environment and not live and ignore other reasons the CG would have taken longer drawing a diagram in draw.io:
- Draw.io is a general-purpose diagram editor: elements have to be manually positioned, connections are manually drawn, the layout has to happen manually, and text for labels has to be copied over.
- This is qualitatively different from annotating design pattern roles directly on syntax elements in an editor and automatically generating live a visualization from it.
- Yes, this is how UML diagrams are typically created, but it might be inefficient and take longer because of many reasons, not just because the users had to switch to a different application.

All issues have been successfully addressed.

Validity of the findings

I did not have any issues with the validity of the findings.

Additional comments

The authors addressed all my concerns, and I would propose accepting it now.

Cite this review as

Reviewer 3 ·

Basic reporting

I found this manuscript written well, with sufficient and relevant references.

Experimental design

The research questions are well-defined. I and other reviewers have already communicated the issues in this section to the authors, and the authors have adequately addressed most of the concerns in the revised manuscript.

Validity of the findings

The findings are supported by results.

Cite this review as

---

## Round 0.2 · accepted · Accept

In the rebuttal letter the authors noted that they can no longer access some of their data. It only affects some of the data and you document this circumstance transparently in the paper, so I find it unfortunate, but still acceptable.

The suggestions made in the minor revision were considered, and I am satisfied with the current version. The manuscript is ready for publication.